# Unraveling the Role of Peroxisome Proliferator-Activated Receptor β/Δ (PPAR β/Δ) in Angiogenesis Associated with Multiple Myeloma

**DOI:** 10.3390/cells12071011

**Published:** 2023-03-25

**Authors:** Patrizia Leone, Antonio Giovanni Solimando, Marcella Prete, Eleonora Malerba, Nicola Susca, Afshin Derakhshani, Paolo Ditonno, Carolina Terragna, Michele Cavo, Nicola Silvestris, Vito Racanelli

**Affiliations:** 1Department of Interdisciplinary Medicine, School of Medicine, ‘Aldo Moro’ University of Bari, 70124 Bari, Italy; 2Department of Precision and Regenerative Medicine—Ionian Pole, School of Medicine, ‘Aldo Moro’ University of Bari, 70124 Bari, Italy; 3IRCCS Istituto Tumori ‘Giovanni Paolo II’ of Bari, 70124 Bari, Italy; 4McCaig Institute for Bone and Joint Health, University of Calgary, Calgary, AB T2N 1N4, Canada; 5Department of Microbiology, Immunology, and Infectious Diseases, Cumming School of Medicine, University of Calgary, Calgary, AB T2N 1N4, Canada; 6Hematology Unit, IRCCS Istituto Tumori ‘Giovanni Paolo II’ of Bari, 70124 Bari, Italy; 7Seràgnoli’ Institute of Hematology, Bologna University School of Medicine, 40126 Bologna, Italy; 8Medical Oncology Unit, Department of Human Pathology “G. Barresi”, University of Messina, 98122 Messina, Italy

**Keywords:** multiple myeloma, MGUS, angiogenesis, tumor progression, PPAR β/δ

## Abstract

Growing evidence suggests a role for peroxisome proliferator-activated receptor β/δ (PPAR β/δ) in the angiogenesis, growth, and metastasis of solid tumors, but little is known about its role in multiple myeloma (MM). Angiogenesis in the bone marrow (BM) is characteristic of disease transition from monoclonal gammopathy of undetermined significance (MGUS) to MM. We examined the expression and function of PPAR β/δ in endothelial cells (EC) from the BM of MGUS (MGEC) and MM (MMEC) patients and showed that PPAR β/δ was expressed at higher levels in MMEC than in MGEC and that the overexpression depended on myeloma plasma cells. The interaction between myeloma plasma cells and MMEC promoted the release of the PPAR β/δ ligand prostaglandin I2 (PGI2) by MMEC, leading to the activation of PPAR β/δ. We also demonstrated that PPAR β/δ was a strong stimulator of angiogenesis in vitro and that PPAR β/δ inhibition by a specific antagonist greatly impaired the angiogenic functions of MMEC. These findings define PGI2-PPAR β/δ signaling in EC as a potential target of anti-angiogenic therapy. They also sustain the use of PPAR β/δ inhibitors in association with conventional drugs as a new therapeutic approach in MM.

## 1. Introduction

Multiple myeloma (MM) is an incurable hematological cancer characterized by transformed plasma cells which proliferate in an uncontrolled way within the bone marrow (BM) [1]. According to a multistep development model, MM progresses from a stable asymptomatic premalignant condition, termed monoclonal gammopathy of undetermined significance (MGUS), to fulminant MM [2]. In most patients with MGUS, the disease remains stable throughout their life, with only a small proportion of patients (~1% per year) developing MM [3]. The evolution and progression of MM and its precursors over time are poorly understood, although many studies have shown that the MGUS to MM progression is stimulated by direct cell interactions and by soluble signaling molecules that mediate cell–cell communication within the BM microenvironment, thus promoting myeloma plasma cell survival and proliferation [2,4,5,6,7]. A crucial interaction is that between myeloma plasma cells and endothelial cells (EC), which form new blood vessels in a process known as angiogenesis. Angiogenesis is a key event during the early phase of the MGUS to MM transition [8]. Pro-angiogenic factors and a variety of other molecules act in concert to generate a vascular network that sustains the high proliferative rate of myeloma plasma cells [9].

The peroxisome proliferator-activated receptor β/δ (PPAR β/δ) plays a key role in both physiological and pathological angiogenesis (reviewed in [10]). One of the target genes of PPAR β/δ encodes for angiopoietin-like 4 (ANGPTL4), a secretory protein involved in angiogenesis, cancer progression, and metastasis [11]. The activation of PPAR β/δ by their natural ligands, such as prostacyclin (prostaglandin I2, PGI2) [12], and selective synthetic agonists such as GW501516, induces angiogenesis and EC proliferation, inhibits EC apoptosis in vitro, and stimulates the proliferation of human breast and prostate cancer cell lines [13,14,15]. PGI2 is very unstable and is spontaneously converted to 6-keto PGF1α in vivo [16]. The activation of PPAR β/δ and, to a much lesser extent, of PPARα [17,18] by PGI2 accounts for the vascular effects of PGI2 [19].

As a ligand-inducible transcription factor belonging to the nuclear hormone receptor superfamily, PPAR β/δ is involved in the control of diverse physiological processes [20,21]. Accumulating evidence suggests that the function of PPAR β/δ is context-dependent, changing from a healthy to a diseased status and from one disease to another. High levels of PPAR β/δ have been found in many human cancers, including colon, breast, and gastric cancers [17,22,23,24,25]. However, the exact role of PPAR β/δ in carcinogenesis is unclear [21,26,27,28].

So far, a very few studies have analyzed the role of PPAR β/δ in MM. Whereas two old studies highlighted its ability to decrease myeloma cell growth through the down-regulation of NFκB activity [29,30], two very recent papers demonstrated PPAR β/δ upregulation in the BM CD138+ plasma cells and BM CD138− microenvironment cells in patients with newly diagnosed MM compared with those in normal BM controls, and a higher PPAR β/δ expression was associated with worse progression-free survival (PFS) and overall survival (OS) [31,32]. Moreover, Wu et al. investigated the drug–drug interactions between immunomodulatory agents (IMiDs) and PPAR agonists in patients with MM, demonstrating opposite metabolic effects of these molecules in MM cells [32]. Through a retrospective study, they assessed that the coadministration of a PPAR agonist with IMiDs was associated with worse PFS and OS in MM patients with co-existing type II diabetes and/or dyslipidemia [32]. However, neither study investigated the link between PPAR β/δ and angiogenesis in MM.

In this study, we analyzed PPAR β/δ expression and activity in BM EC from MGUS and MM patients in order to test the hypothesis that the PGI2-PPAR β/δ signaling pathway is involved in BM angiogenesis and MM progression.

## 2. Materials and Methods

### 2.1. Biological Samples

The International Myeloma Working Group criteria [33,34] were used to classify patients as having MGUS (*n* = 20) or symptomatic newly diagnosed MM (*n* = 20). The clinical characteristics of the enrolled patients are summarized in Table 1. Heparinized BM aspirates were collected following informed consent obtained by the University of Bari Medical School Ethics Committee (# 5145). All studies were performed following reception of a written informed consent from the patients and in accordance with the Declaration of Helsinki and the Good Clinical Practice Guidelines of the Italian Ministry of Health.

### 2.2. Cell Isolation and Culture

Bone marrow mononuclear cells (BMMC) obtained by density-gradient centrifugation were cultured in Roswell Park Memorial Institute (RPMI)-1640 medium supplemented with 10% heat-inactivated fetal bovine serum (FBS). For primary EC immunomagnetic isolation from MGUS (MGEC) and MM patients (MMEC), BMMC were expanded in Dulbecco’s modified Eagle’s medium (DMEM) containing 20% FBS, 2 mM L-glutamine, 100 U of penicillin/mL, and 100 µg of streptomycin/mL (Sigma-Aldrich, Buchs, Switzerland; culture medium), as previously described [35]. Briefly, adherent BMMC were grown for 3 weeks in the culture medium and then used for automated magnetic separation of EC using anti-CD31 microbeads (Miltenyi Biotec, Bergisch Gladbach, Germany). Purified EC were expanded for six passages in fibronectin-coated culture dishes (BD Falcon) in endothelial basal medium (EBM-2, Lonza, Basilea, Switzerland) supplemented with 5% FBS. Cell purity (>95%) was assessed by flow cytometry of immunostained cells using a BD Accuri™ C6 flow cytometer (Becton Dickinson-BD, San Jose, CA, USA).

The BMMC supernatants were recovered after 7 days of culture and employed as conditioned medium (MGUS BM medium or MM BM medium) for the treatment of MMEC for 24 h and for assessing 6-keto PGF1α concentrations by ELISA.

For the co-culture assays, MMEC (4 × 10^5^ cells/100 mm Petri dish) were cultured alone or with the MM cell line RPMI 8226 (American Type Culture Collection) cells at 1:1 and 1:10 ratios, separated (indirect co-culture) or not (direct co-culture) by a Transwell membrane (0.4 µm pore size; Corning, New York, NY, USA). After 24 h, the supernatants were gathered and stored at −80 °C for ELISA, and MMEC were used for western blotting, quantitative real-time PCR, and analysis of PPAR β/δ transcriptional activity. For co-cultures without a Transwell membrane, MMEC were immunomagnetically separated from RPMI 8226 cells using CD31 microbeads.

### 2.3. Functional Assays

For the functional assays, MMEC were treated with 10 µM GW501516 (PPAR β/δ agonist) and 1 µM GSK3787 (PPAR β/δ antagonist) (both from Sigma Aldrich). Matrigel matrix was acquired from Becton Dickinson-BD Biosciences. 

For the in vitro spreading assays, MMEC (4 × 10^3^ cells/well) were plated on fibronectin-coated (10 mg/mL) 96-well plates in serum-free medium alone (control) or containing GW501516. After 1 h, the cells were fixed with 4% paraformaldehyde, stained with crystal violet (both from Sigma-Aldrich), and observed with an EVOS microscope. Round cells were annotated as unspread cells, and cells with visible cytoplasm surrounding the nuclei and with membrane protrusions as spread cells. The spreading capacity was quantified using the VICTOR X reader.

Wound-healing assays were carried out as previously described [36]. Briefly, MMEC were cultured until they reached confluence on fibronectin-coated 6-well plates, and a cell-free area (wound) was created by scratching the cell monolayer with a P200 pipette tip. The cells were grown in serum-free medium alone (control) or in medium containing GW501516 or GSK3787. Afterward, they were fixed with 4% paraformaldehyde and stained with crystal violet (both from Sigma-Aldrich). Cell migration and wound closure were tracked using an EVOS microscope (Thermo Fisher Scientific, Waltham, MA, USA). The migrating MMEC were counted into three different fields of the wound.

For the in vitro angiogenesis experiments, MMEC were cultured on Matrigel-coated 48-well plates in serum-free medium without (control) or with GW501516 for 4, 12, and 30 h. In some experiments, serum-free medium supplemented with GSK3787 was added to the cells for 20 h.

Photomicrographs of skeletonized meshes, as indicators of angiogenesis, in three randomly chosen fields were taken through an EVOS microscope. Topological parameters (mesh areas, length, and branching points) were evaluated using a computerized image analyzer as previously described [36]. The data were normalized to control values.

### 2.4. Western Blot

Following MGEC and MMEC lysis using RIPA lysis and extraction buffer (Thermo Fisher Scientific), total protein extracts (30 µg) were subjected to SDS-PAGE and transferred to a nitrocellulose membrane (Bio-Rad, Hercules, CA, USA). Immunoblots were performed overnight using the following antibodies: anti-PPAR β/δ antibody (Thermo Fisher Scientific), anti-β-actin antibody (Sigma-Aldrich), and mouse and rabbit horseradish peroxidase-conjugated anti-IgG (Bio-Rad). The immunoreactive bands were detected by SuperSignal West Femto Maximum Sensitivity Substrate (Thermo Fisher Scientific) using the Gel Logic 1500 imaging system (Eastman Kodak, New York, NY, USA). Densitometry analysis of band intensity was performed using Kodak molecular imaging software 5.0 (Kodak, New York, NY, USA). The results are reported as the relative density.

### 2.5. Real-Time PCR

RNA was purified from MGEC and MMEC using the RNeasy micro-kit (Qiagen, Hilden, Germany) and reverse-transcribed to complementary DNA (cDNA) using the first-strand cDNA synthesis kit (Thermo Fisher Scientific). The mRNA levels of PPAR β/δ, ANGPTL4, elastin, collagen 3α, fibronectin, and glyceraldehyde 3-phosphate dehydrogenase (GAPDH) were assessed in triplicate by real-time PCR using TaqMan kits (Applied Biosystems (Waltham, MA, USA) assay IDs: Hs00987008_m1, Hs00211522_m1, Hs00355783_m1, Hs00915125_m1, Hs01549976_m1 and Hs03929097_g1) and the StepOne real-time PCR system (Applied Biosystems). The comparative Ct method, with GAPDH as the reference gene, and the 2^−ΔΔCt^ formula were used for the relative quantification of mRNA levels.

### 2.6. ELISA

A 6-keto PGF1α ELISA kit (Cayman Chemical, Ann Arbor, MI, USA) was used according to the manufacturer’s instructions.

### 2.7. PPAR β/δ Transcriptional Activity

Nuclear extracts were obtained from MMEC cells using a nuclear extraction kit (Abcam), and the transcriptional activity of PPAR β/δ was assessed in a PPAR β/δ transcription factor assay (Abcam) performed according to the manufacturer’s instructions.

### 2.8. In Vivo Experiments

We inoculated 2 × 10^5^ RPMI-8226 cells suspended in phosphate-buffered saline into the tibial of ten female 6- to 8-week-old non-obese diabetic/severe combined immunodeficiency mice (NOD.CB17-Prkdcscid/NCrHsd; Envigo, Huntingdon, UK). After 7 days, GSK3787 (10 mg/kg) was administered to the mice once every second day for 6 days [23]. The mice were sacrificed on day 14. Formalin-fixed tumor tissue samples were embedded in paraffin, cut in 5 µm sections, and treated for immunohistochemical analysis. The DAKO Advance system was used for detection, according to the manufacturer’s protocol. Anti-CD31 (ab124432, Abcam, Cambridge, UK) and -Ki67 (LS-C175347, LifeSpan BioSciences, Seattle, WA, USA) antibodies were used for staining the BM sections, according to the manufacturer’s instructions. Two pathologists independently analyzed the immunohistochemical images in a blind fashion. For the quantification of positively stained cells, each tumor was examined on five different slides, and each slide was explored in five fields; the positive cells were reported as a percentage of all cells.

The mice were housed according to the protocol approved by the Institutional Animal Care and Use Committee of the University Medical School of Bari (license n. 846/2017PR).

### 2.9. Survival Analysis of the GSE9782 Multiple Myeloma Dataset

The gene expression data and clinical features of 264 patients with relapsed MM were obtained from the GSE9782 dataset (Mulligan et al. study) [37]. For Kaplan–Meier plotter analysis, the patients were divided into two groups, PPAR β/δ^low^ and PPAR β/δ^high^, based on the median PPAR β/δ mRNA level. To assess differences in overall survival, the log rank test was used.

### 2.10. Statistical Analysis

GraphPad Prism5 software was used for statistical analyses. Given that many of the data were not distributed normally, nonparametric statistics were applied with significance set to *p* value < 0.05. Statistical tests included the Mann–Whitney U test for comparisons of groups, the Wilcoxon signed-rank test for comparisons of matched samples, and Spearman’s rank test for correlations.

## 3. Results

The role of PPAR β/δ in BM angiogenesis and MM progression was investigated by analyzing its expression and activity in BM EC from 20 MM and 20 MGUS patients. Primary MGEC and MMEC from BM samples were immunomagnetically purified to >95% purity. The measurements of PPAR β/δ mRNA and protein levels showed that both were significantly lower in MGEC than in MMEC (*p* = 0.0010 and *p* = 0.0002, respectively) (Figure 1A,B).

Whether PPAR β/δ expression is affected by the BM microenvironment was examined by treating MMEC with the conditioned culture medium of BMMC obtained from MGUS and MM patients and then measuring PPAR β/δ protein levels before and after treatment. The PPAR β/δ levels were not influenced by the conditioned medium from MGUS BMMC (Figure 2A), while they increased in MMEC treated with the conditioned medium from MM BMMC (*p* = 0.0079) (Figure 2B). The association of increased PPAR β/δ levels with increased myeloma plasma cells and thus with tumor progression was determined by simulating EC–plasma cell interactions during tumor growth in an experimental co-culture system. Specifically, MMEC were cultured in the presence of increasing numbers of RPMI 8226 myeloma cells, either in contact with those cells (direct co-culture) or separated by a Transwell membrane (indirect co-culture). PPAR β/δ protein and mRNA levels were significantly higher in both co-culture settings and increased in proportion to the amount of RPMI 8266 cells in the culture (MMEC/RPMI 8226 ratios of 1:1 and 1:10) (Figure 2C–E).

The association of the increase in PPAR β/δ expression observed in the co-culture experiment with the enhancement of transcriptional activity was then assessed by measuring the PPAR β/δ DNA binding activity in nuclear extracts of MMEC after indirect and direct culture with RPMI 8226 myeloma cells at ratios of 1:1 and 1:10 (MMEC/RPMI 8226). PPAR β/δ transcriptional activity was detected in both co-culture conditions at the two ratios tested, but to a greater extent in the direct co-culture at the higher cell ratio (*p* = 0.0022) (Figure 2F,G). Given that the activity of a transcription factor modulates the expression of its target genes, the expression of ANGPTL4 was measured as well. A significant up-regulation of ANGPTL4 mRNA expression levels in MMEC was observed in both direct and indirect co-cultures at the two ratios tested, with a larger increase in the direct co-culture at a cell ratio of 1:10 (*p* = 0.0006) (Figure 2H).

Overall, these results indicated that the expression of PPAR β/δ was higher in BM EC from MM patients than in those from MGUS patients. They also imply that the expression and activity of PPAR β/δ in BM EC from MM but not MGUS patients were affected by soluble factors whose release depended on myeloma plasma cells. 

As PGI2 is an endogenous activating ligand of PPAR β/δ, its ability to activate this receptor in MMEC was explored by examining the PGI2 levels in the culture media of MGEC and MMEC. Because of the high instability of PGI2, such that it rapidly undergoes spontaneous transformation to 6-keto PGF1α, the levels of the latter were measured as a proxy of PGI2. The results showed significantly greater levels in the medium from MMEC than in that from MGEC (*p* = 0.0006) (Figure 3A). Whether myeloma plasma cells modulated PGI2 (6-keto PGF1α) production by MMEC was examined by measuring the PGI2 (6-keto PGF1α) concentration in supernatants from co-cultures with RPMI 8226 cells. Those experiments showed that PGI2 was released by MMEC, but its concentration was significantly enhanced in the presence of increasing numbers of myeloma cells, both in indirect and, especially, in direct co-cultures (140 pg/mL vs. 80 pg/mL) (Figure 3B,C). Moreover, endogenous PGI2 production correlated positively with PPAR β/δ transcriptional activity (r = 0.7568) (Figure 3D). The levels of PGI2 were also found to be higher in the serum of MM patients than in the serum of MGUS patients (*p* = 0.0007) (Figure 3E). These results showed that the myeloma plasma cells stimulated MMEC to produce PGI2, which then activated PPAR β/δ.

Whether PPAR β/δ activation influenced the in vitro MMEC angiogenic functions was explored as well, by treating MMEC with the synthetic PPAR-β/δ-specific agonist GW501516 and then assessing angiogenic functions in vitro (Figure 4). The chosen concentration of GW501516 (10 nM) was based on a previous concentration–response evaluation over a range of 1–50 nM.

In a cell-spreading assay, treatment with the PPAR β/δ agonist GW501516 greatly stimulated the formation of cellular protrusions and changes in MMEC shape (Figure 4A). The cell-covered area significantly increased due to new cell–matrix adhesions (Figure 4B). In a wound-healing assay, 12 h after the monolayers were scratched, the migration of GW501516-treated MMEC was faster than that of untreated cells, resulting in a more rapid wound closure (Figure 4C). This finding was corroborated by the number of MMEC that had migrated into the wound (Figure 4D), as cell migration significantly increased, with 98% of GW501516-treated cells having migrated after 12 h compared to 40% of control cells (Figure 4C,D).

Similar results were obtained in an in vitro angiogenesis assay in which MMEC were cultured on Matrigel-coated plates in medium with or without GW501516 (Figure 4E). After 4 h, the control sample showed an organization at the early stage, with cell clusters disseminated on Matrigel, whereas the GW501516-treatment was accompanied by the formation of capillary-like structures. The complexity of the vascular network enhanced with an increasing treatment time (12 h and 30 h), as shown by substantial increments in topological parameters (Figure 4E). By 12 h, GW501516-treated MMEC had created a stabilized vascular network characterized by well-defined areas with elongated, juxtaposed cells. This vascular network remained relatively stable even at 30 h, whereas in the control it almost completely degenerated (Figure 4E).

The mechanism by which GW501516 treatment may stabilize the vessel structure was investigated by examining whether PPAR β/δ activation indirectly influenced extracellular matrix remodeling, as determined based on the mRNA levels of elastin, collagen 3α, and fibronectin in MMEC grown in the presence or absence of 10 nM GW501516 for 12 h. All three analyzed extracellular matrix proteins increased at the mRNA level after GW501516 treatment (Figure 5A). In addition, in MMEC treated with GW501516, the mRNA levels of ANGPTL4 were significantly enhanced (Figure 5B).

Overall, these results indicated that PPAR β/δ activation strongly stimulated MMEC angiogenic functions, conferring on the cells a rapid and high pro-angiogenic capacity and the ability to stabilize the vascular network over time.

To firmly establish the PPAR β/δ-dependent angiogenic effect, MMEC were cultured in the presence or absence of the specific inhibitor GSK3787, after which in vitro angiogenesis assays were performed. In a wound-healing assay, GSK3787 treatment greatly reduced MMEC migration (*p* = 0.0022) (Figure 6A,B). In an in vitro angiogenesis assay, capillary network formation was impaired in GSK3787-treated MMEC, evidenced by broken connections and isolated cells (Figure 6C). Moreover, all three topological parameters were significantly lower than in the control (untreated MMEC) (Figure 6C). Finally, GSK3787 treatment significantly down-regulated the mRNA levels of elastin, collagen 3α, fibronectin, and ANGPTL4 in MMEC (Figure 6D).

To investigate whether PPAR β/δ inhibition may affect in vivo angiogenesis and in turn MM cell growth, we used a xenograft MM mouse model and analyzed bone specimens after GSK3787 treatment. Immunohistochemical analyses revealed that PPAR β/δ inhibition greatly decreased the percentage of EC staining positively for CD31, a marker of microvessel density, as well as the percentage of Ki-67-positive cells, a marker of proliferation (Figure 7). Altogether, these results demonstrated that PPAR β/δ inhibition reduced MM angiogenesis and cell proliferation.

Finally, to evaluate the clinical implications of PPAR β/δ overexpression, we investigated the association between PPAR β/δ expression levels and overall survival in 264 MM patients enrolled in the Mulligan et al. study [37] (GSE9782 dataset). To this aim, we compared patients below the median of PPAR β/δ expression (PPAR β/δ^low^) with those with the highest levels (over the median, PPAR β/δ^high^). Kaplan–Meier survival analyses indicated shorter OS (*p* = 0.035) for patients of the PPAR β/δ^high^ subgroup (Figure 8).

## 4. Discussion

This study identified the PGI2-PPAR β/δ signaling pathway as a promoter of angiogenesis in MM. PPAR β/δ expression and activity were higher in EC from MM patients (MMEC) than in EC from MGUS patients (MGEC), and interestingly, PPAR β/δ activity was regulated by the PGI2 concentration in the BM milieu. PGI2 is an inflammatory mediator mainly produced by EC [38,39] that is involved in blood vessel formation and angiogenesis [40]. Our study suggests that MMEC make use of an autocrine loop that is intensified by myeloma plasma cells. The latter stimulate PGI2 release by EC, which respond to increasing concentrations of PGI2 in the surrounding milieu by augmenting PPAR β/δ activity. According to this sequence of events, during the transition from MGUS to MM, and thus from the avascular to the vascular phase, the proliferation of myeloma cells together with changes in the BM microenvironment induce the up-regulation of PPAR β/δ and PGI2. Nevertheless, we cannot exclude the contribution of other BM cells such as vascular smooth muscle cells and fibroblasts in PGI2 production, even if in much smaller quantities [41].

PPAR β/δ is significantly up-regulated in many solid human cancers, including lung, prostate, breast, endometrial adenocarcinoma, gastric, and pancreatic cancer [17,22,24,25,42], and the expression of these receptors strongly correlates with advanced pathological and clinical parameters. For instance, in pancreatic cancer, PPAR β/δ acts as critical “hub node” transcription factor that regulates the tumor angiogenic switch [42]. In patients with pancreatic cancer, PPAR β/δ expression and activation indicate a highly angiogenic phenotype and strongly correlate with the tumor stage as well as with an increased risk of tumor relapse or distant metastasis. These observations are consistent with the critical involvement of PPAR β/δ in tumor angiogenesis and progression [42] and corroborate our findings that a high PPAR β/δ expression is associated with poor overall survival in patients with MM. They also support our results that PPAR β/δ activation in MMEC amplifies the pro-angiogenic capacity of these cells in the early phase of vessel formation and then stabilizes the vascular network over time. Moreover, the up-regulation of PPAR β/δ was associated with the increased expression of both ANGPTL4, a PPAR β/δ target gene involved in angiogenesis and tumor development [43,44,45], and the extracellular matrix proteins elastin, collagen 3α, and fibronectin. Angiogenesis thus appears to be a multistep process involving a widely interactive network made up of multiple proteins and signaling molecules, that includes PPAR β/δ as direct or indirect modulators of the intensive crosstalk promoting angiogenesis. Finally, our finding that PPAR β/δ inhibition strongly impaired angiogenesis points to PPAR β/δ as a potential drug target in the treatment of MM.

So far, increased PPAR β/δ expression has been linked to the pathogenesis of hematological malignancies only in chronic lymphocytic leukemia (CLL) [46,47]. In B lymphoma cell lines and primary CLL cells, PPAR β/δ modulates cholesterol metabolism and cytokine signaling [46], reduces oxidative stress, and increases the metabolic efficiency, thus promoting cell survival under energetically stressful conditions such as hypoxia and chemotherapy [47]. Whether PPAR β/δ has a similar role in myeloma cells with great lipid accumulation after treatment should be explored.

The involvement of the recently described EGFR/HSP90/PPAR β/δ pathway in tumor cell metabolism, proliferation, and chemoresistance suggests a wider role of PPAR β/δ, one that includes cancer development and the resistance to therapy [48]. In response to epidermal growth factor (EGF) and its activated receptor (EGFR), PPAR β/δ protein levels and stability are increased via the recruitment of HSP90 (heat shock protein 90) in cancer cell lines [48]. We previously demonstrated that EGFR is highly expressed on BM EC from MM patients and that binding of the ligand HB-EGF triggers a signaling pathway involved in MM angiogenesis and progression [36].

## 5. Conclusions

The findings presented in this study propose PPAR β/δ as a key inducer of MM angiogenesis and development. To our knowledge, this is the first report to definitely associate PPAR β/δ with the angiogenic switch underlying the MGUS-to-MM transition. Moreover, the inhibition of PPAR β/δ by PPAR antagonists resulted in decreased angiogenesis and cell proliferation, suggesting the therapeutic targeting of PPAR, alone or in combination with conventional MM immunotherapies. Additional studies are needed to define drug–drug interactions between PPAR β/δ antagonists and anti-tumor agents such as IMiDs.

## Figures and Tables

**Figure 1 cells-12-01011-f001:**
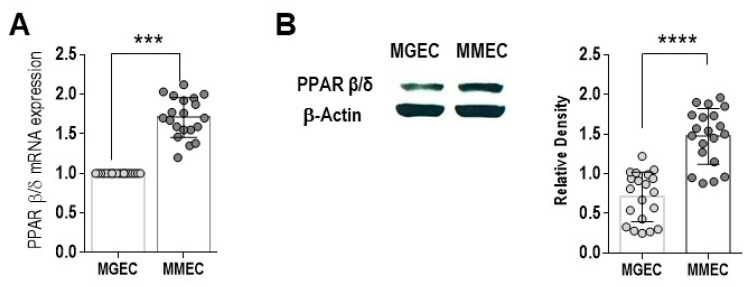
PPAR β/δ expression in bone marrow endothelial cells from MGUS (MGEC) and MM (MMEC) patients. (**A**) Relative mRNA levels of PPAR β/δ in MGEC and MMEC. The values were normalized to those of MGEC and determined by real time-PCR and the 2^−ΔΔCt^ method. (**B**) Western blot and densitometric analysis of the basal expression of PPAR β/δ protein in MGEC and MMEC lysates, normalized to β-actin. The presented values are the mean ± SD from 20 MGUS and 20 MM patients. *** *p* ≤ 0.001 and **** *p* ≤ 0.0001, Mann–Whitney test.

**Figure 2 cells-12-01011-f002:**
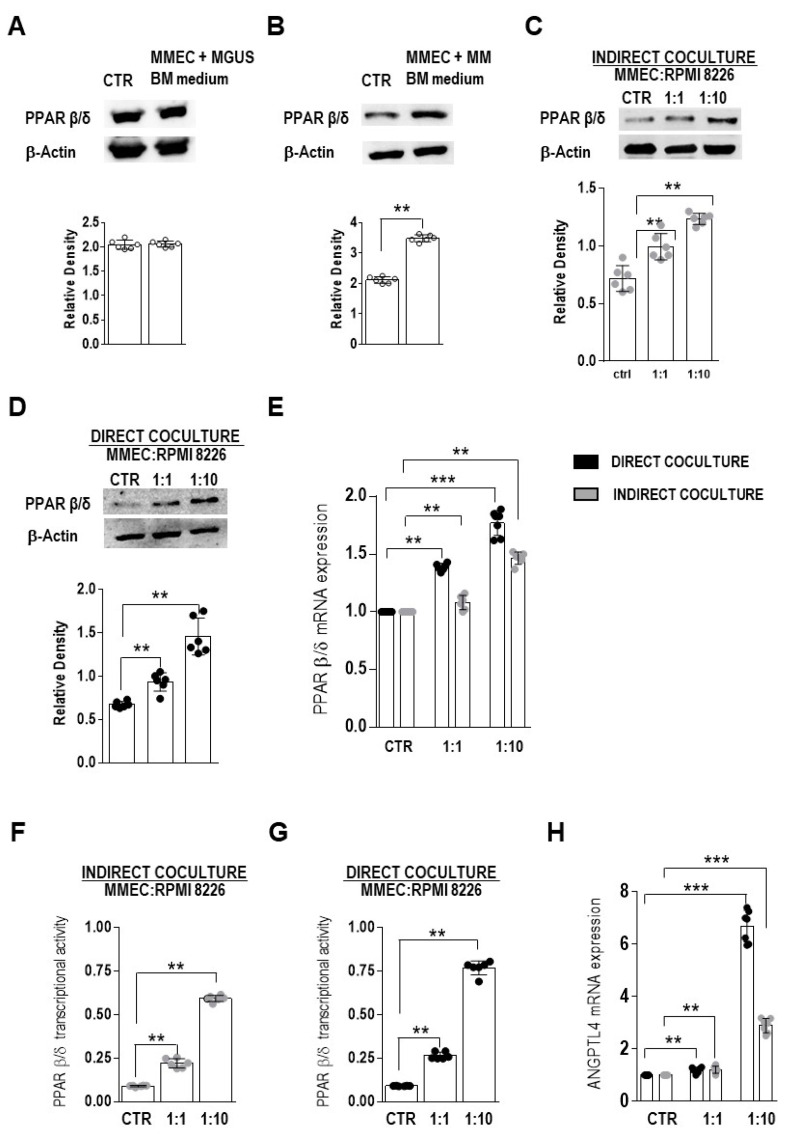
Impact of the bone marrow (BM) microenvironment on PPAR β/δ expression in BM endothelial cells from MM patients (MMEC). (**A**,**B**) MMEC were cultured for 24 h in culture medium (CTR) or culture medium conditioned by bone marrow mononuclear cells (BMMC) from (**A**) MGUS patients (MGUS BM medium) or (**B**) MM patients (MM BM medium). Western blots and densitometric analyses of PPAR β/δ expression in MMEC lysates normalized to β-actin. (**C**–**H**) MMEC were cultured for 24 h, either alone (CTR) or with RPMI 8226 cells at 1:1 and 1:10 cell ratios. The cells were separated (indirect co-culture) or not (direct co-culture) by Transwell membrane inserts. (**C**,**D**) Western blots and densitometric analysis of the expression of PPAR β/δ in MMEC (normalized to β-actin) under different culture conditions. (**E**) Relative PPAR β/δ mRNA levels in MMEC under different culture conditions. The values were normalized to those of the control and determined by real time-PCR and the 2^−ΔΔCt^ method. (**F**,**G**) PPAR β/δ transcriptional activity in MMEC under different culture conditions. (**H**) Relative ANGPTL4 mRNA levels in MMEC cultured alone (CTR) or with RPMI 8226 cells, either together (black histograms) or separated by a Transwell membrane (gray histograms). Values were normalized to those of the control and determined by real time-PCR and the 2^−ΔΔCt^ method. All plots report data from six patient samples tested in triplicate. The values are expressed as the mean ± SD. ** *p* ≤ 0.01 *** and *p* ≤ 0.001, Mann–Whitney test.

**Figure 3 cells-12-01011-f003:**
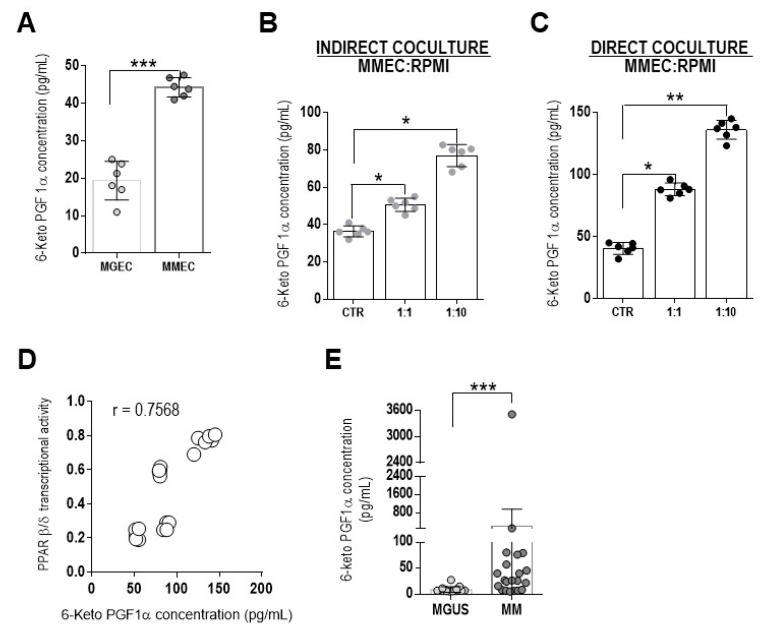
PPAR β/δ soluble endogenous activating ligand 6-keto prostaglandin F1α (6-keto-PGF1α) concentration in cultured cells and serum from MGUS and MM patients. (**A**–**C**) ELISA determination of the 6-keto-PGF1α concentration in supernatants from (**A**) MGEC and MMEC, (**B**,**C**) MMEC cultured for 24 h, alone (CTR) or with RPMI 8226 cells at 1:1 and 1:10 cell ratios and separated (**B**) or not (**C**) by Transwell inserts. The values are expressed as the mean ± SD. * *p* < 0.05, ** *p* ≤ 0.01 and *** *p* ≤ 0.001, Mann–Whitney test. (**D**) Correlation between the 6-keto-PGF1α concentration and PPAR β/δ transcriptional activity in MMEC according to a Spearman’s rank test. (**E**) ELISA determination of the 6-keto-PGF1α concentration in the sera from MGUS and MM patients. The values are expressed as the mean ± SD. *** *p* ≤ 0.001, Mann–Whitney test.

**Figure 4 cells-12-01011-f004:**
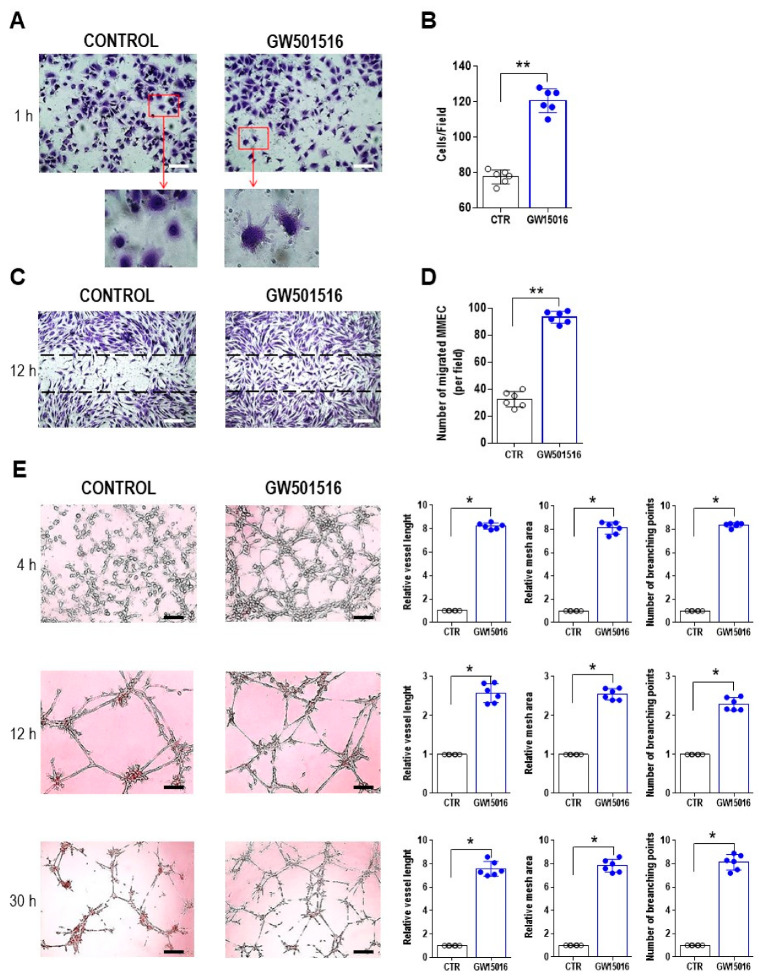
Stimulation of in vitro MMEC spreading, migration, and angiogenesis by a PPAR β/δ agonist. MMEC were cultured in serum-free medium alone (control) or in containing 10 nM GW501516 for different times. (**A**,**B**) Cell spreading assay. (**A**) Representative photomicrographs at 1 h; the cells were harvested, stained with crystal violet, and observed using an EVOS microscope. Round cells were considered unspread, and cells with a visible cytoplasm surrounding the nuclei and with membrane protrusions as spread. 200×; Scale bar, 50 μm. (**B**) Plot showing the percentage of spread cells. (**C**,**D**) Wound-healing assay. (**C**) Representative micrographs 12 h after confluent monolayers were scratched (dotted lines define the wound area). 200×; Scale bar, 50 μm. (**D**) Number of migrating cells in each wound of (**C**). (**E**) Matrigel angiogenesis assay. Representative micrographs at 4, 12, and 30 h of treatment revealing newly formed capillary networks on a Matrigel layer. The cell angiogenic behavior was evaluated by a topological analysis. 200×; Scale bar, 50 μm. Data are the mean ± SD of six independent experiments and were normalized to the control. * *p* < 0.05, ** *p* ≤ 0.01, Mann–Whitney test.

**Figure 5 cells-12-01011-f005:**
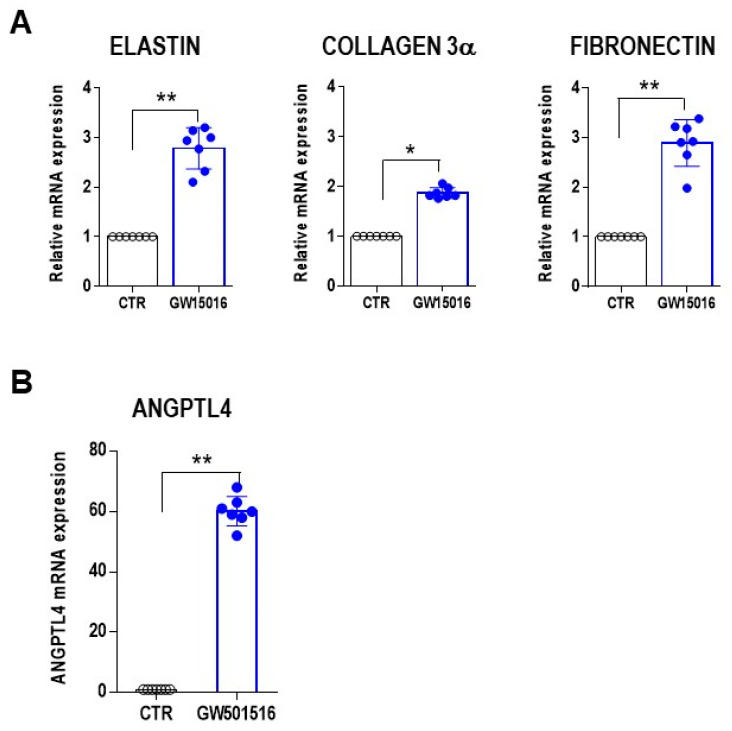
Promotion of extracellular matrix remodeling and ANGPTL4 expression in MMEC treated with a PPAR β/δ agonist. (**A**,**B**) Relative mRNA levels of (**A**) elastin, collagen 3α, fibronectin, and (**B**) ANGPTL4 in MMEC cultured in serum-free medium alone (CTR) or containing 10 nM GW501516 for 12 h. The values were normalized to those of the control and were determined by real-time PCR and the 2^−ΔΔCt^ method based on seven independent experiments. * *p* < 0.05, ** *p* ≤ 0.01, Mann–Whitney test.

**Figure 6 cells-12-01011-f006:**
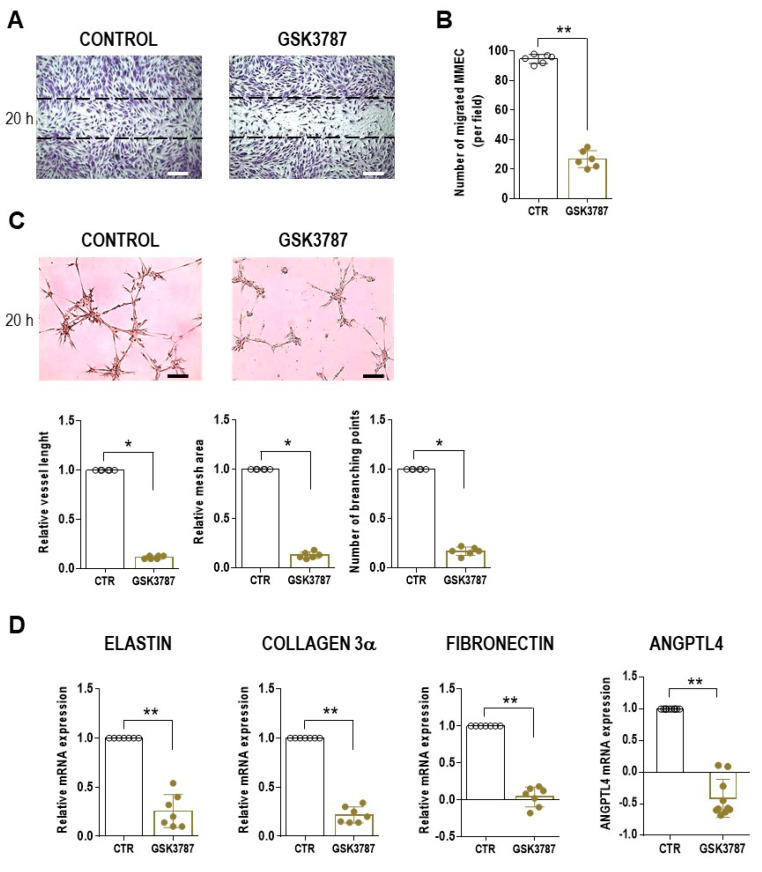
Inhibition of in vitro MMEC migration and angiogenesis by a PPAR β/δ antagonist. MMEC were cultured in serum-free medium alone (control) or containing 1 µM GSK3787 for 20 h. (**A**) Confluent monolayers of MMEC were subjected to a wound-healing assay. Photographs were taken 24 h after the wound was made. Dotted lines define the wound area. (**B**) Number of migrating cells in each wound of (**A**). (**C**) Matrigel angiogenesis assay. Representative micrographs of MMEC seeded on Matrigel and topological analysis. 200×; Scale bar, 50 μm. Data are the mean ± SD of six independent experiments and were normalized to the control. * *p* < 0.05, ** *p* ≤ 0.01, Mann-Whitney test. (**D**) Relative mRNA levels of elastin, collagen 3α, fibronectin, and ANGPTL4 in MMEC. The values were normalized to those of the control and determined by real time-PCR and the 2^−ΔΔCt^ method. ** *p* ≤ 0.01, Mann–Whitney test.

**Figure 7 cells-12-01011-f007:**
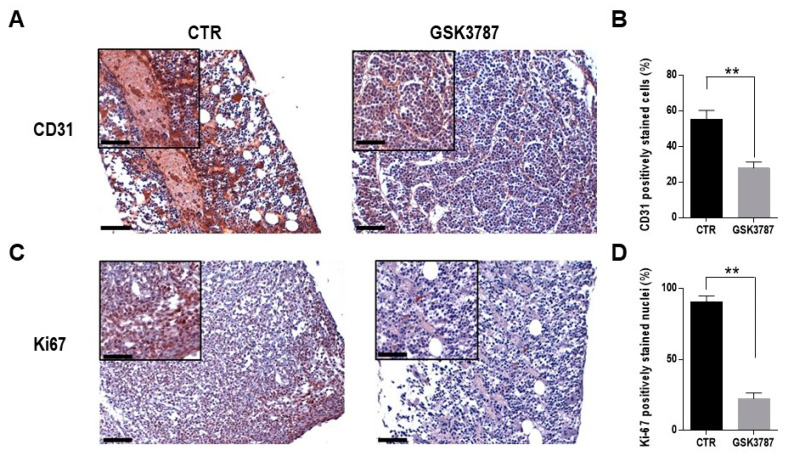
PPAR β/δ inhibition reduces MM angiogenesis. NOD/SCID mice bearing RPMI-8226 intratibial xenografts were treated with GSK3787 (*n* = 5) or vehicle (*n* = 5) for 6 days. (**A**,**B**) Tumor sections stained for CD31 and quantification of CD31-positive murine endothelial cells. (**C**,**D**) Tumor sections stained for Ki-67 and quantification of positively stained nuclei. The histograms show the average results from five slides for each condition and five fields per slide. ** *p* < 0.01, Mann–Whitney U test. Scale bar = 100 µm.

**Figure 8 cells-12-01011-f008:**
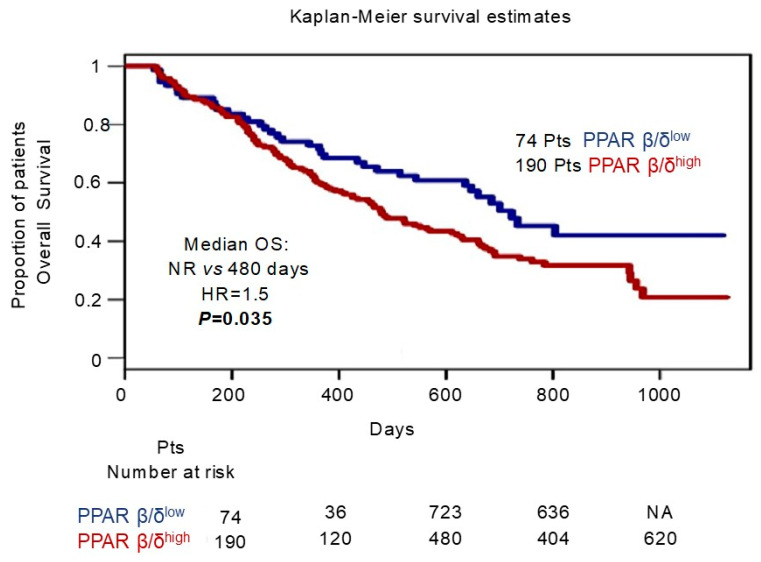
Prognostic significance of PPAR β/δ. OS from the GSE9782 MM dataset. Gene expression profiling of 264 MM patients; PPAR β/δ^low^ (dic = 0) versus PPAR β/δ^high^ (dic = 1) expressors were compared. See text for details.

**Table 1 cells-12-01011-t001:** Clinical characteristics of the patients.

	Variable	N. Patients (%)	Median Values
MGUS			
	Median Age	20/20 (100)	62 years (49–75)
	Male	12/20 (60)	
	Female	8/20 (40)	
	IgG	13/20 (65)	
	IgA	5/20 (25)	
	Light chain	2/20 (10)	
Symptomatic newly diagnosed MM			
	Median Age	20/20 (100)	65.5 years (47–81)
	Male	10/20 (50)	
	Female	10/20 (50)	
	Stage I	5/20 (25)	
	Stage II	8/20 (40)	
	Stage III	7/20 (35)	
	IgG	17/20 (85)	
	IgA	2/20 (10)	
	Light chain	1/20 (5)	

## Data Availability

The data are unavailable due to privacy.

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
