# Peer review of "Unraveling the Role of Peroxisome Proliferator-Activated Receptor β/Δ (PPAR β/Δ) in Angiogenesis Associated with Multiple Myeloma"

_cells, 2023, doi:10.3390/cells12071011_

Round 1

Reviewer 1 Report

Leone et al here report an interesting effect of multiple myeloma (MM) on PPAR β/δ expression in endothelial cells (EC) and its role in regulating the angiogenic activities of MMECs in vitro. Patient samples were used throughout the study, which made the conclusion important for therapeutic applications.

Major comments:

1)    Detail information of patients (general info, diagnose/progression, chemotherapy, and etc) should be provided.

2)    It is unclear how PGI2, as an activating ligand, can affect PPAR β/δ expression.

3)    MMECs may not be the only source of PGI2 in the bone marrow microenvironment. How about other cells? The authors should at least discuss that. Were serum PGI2 levels increased in MM patients?

4)    Can you silence PPAR β/δ in purified MMECs and test if PPAR β/δ siRNA has the same effect, by considering the potential issue of inhibitor specificity?

Minor issues:

1)    The spelling of PPAR β/δ should be consistent between title and text (Upper vs lower case).

2)    The authors are encouraged to convert all bar graphs into scatter plots so that variations among patient sample groups can be visible.

3)    Passages of purified MMECs should be commented.

4)    Fig 1B, there is no need to split the blot (available in the Original Images for Blots/Gels file) into two panels.

5)    Please add scale bars in Figs 4A, C, E, 6A, C.

Author Response

We would like to thank very much the Reviewer for the constructive and helpful comments regarding our manuscript.

Reviewer #1: Detail information of patients (general info, diagnose/progression, chemotherapy, and etc) should be provided.

Action: A Table with clinical characteristics of patients has been added in the section Materials and Methods, pages 2-3, lines 40-46.

Reviewer #1: It is unclear how PGI2, as an activating ligand, can affect PPAR β/δ expression.

Action: Thank you for pointing this out. We agree with the Reviewer and we have, accordingly, eliminated the statement that PGI2 affects PPAR β/δ expression at page 1 line 20, and page 15 lines 8 and 14. What we actually measured was the effect of PGI2 on PPAR β/δ activity.

Reviewer #1: MMECs may not be the only source of PGI2 in the bone marrow microenvironment. How about other cells? The authors should at least discuss that.

Action: We discuss this statement at page15, lines 17-19.

Reviewer #1: Were serum PGI2 levels increased in MM patients?

Action: We thank the Reviewer for the insightful suggestion that improves our manuscript quality. Measurement of PGI2 levels in serum of MGUS and MM patients has been added in the Figure 3E, at page 8, lines 45-46 and page 9.

Reviewer #1: Can you silence PPAR β/δ in purified MMECs and test if PPAR β/δ siRNA has the same effect, by considering the potential issue of inhibitor specificity?

Action: We thank the Reviewer for the comment. However, experiments with PPAR β/δ siRNA are ongoing for another future study. In this paper, to inhibit PPAR β/δ we used GSK3787 as it is a potent, selective and irreversible PPAR β/δ antagonist (pIC50 of 6.6) with no measurable affinity for PPAR α or PPAR γ (pIC50 < 5).

Reviewer #1: The spelling of PPAR β/δ should be consistent between title and text (Upper vs lower case).

Action: Correction has been made.

Reviewer #1: The authors are encouraged to convert all bar graphs into scatter plots so that variations among patient sample groups can be visible.

Action: All bar graphs have been converted into scatter plots.

Reviewer #1: Passages of purified MMECs should be commented.

Action: More information about isolation and expansion of MMECs have been added at page 3, lines 15-24.

Reviewer #1: Fig 1B, there is no need to split the blot (available in the Original Images for Blots/Gels file) into two panels.

Action: The Figure 1B has been changed.

Reviewer #1: Please add scale bars in Figs 4A, C, E, 6A, C.

Action: Scale bars in the Figures 4A, 4C, 4E, 6A and 6C have been added.

Reviewer 2 Report

I have read the manuscript numbered “cells-2115894” and entitled Unraveling the Role of Peroxisome Proliferator-Activated Receptor Β/Δ (Ppar Β/Δ) In Angiogenesis Associated with Multiple Myeloma” with great interest.   The authors aimed to investigate Ppar Β/Δ role in the angiogenesis and whether its expression is variable in MGUS and myeloma.

       The introduction is compact, and the flow is good.

       I believe the authors can mention more on PPAR β/δ and myeloma relationship since there are several recently published articles on the related topic. Immunomodulatory drugs and their effects on PPAR β/δ can be mentioned.

       The method section is well established and detailly written for every step.

       Results were well presented and clear.

       There are several studies investigated Ppar Β/Δ role in tumor angiogenesis as the authors mentioned.  Were there any findings which is inconsistent with the previous literature?

       I believe the limitations of the study should be mentioned. 

Author Response

We would like to thank very much the Reviewer for the constructive and helpful comments regarding our manuscript.

Reviewer #2: I believe the authors can mention more on PPAR β/δ and myeloma relationship since there are several recently published articles on the related topic. Immunomodulatory drugs and their effects on PPAR β/δ can be mentioned.

Action: Taking advantage of two very recent papers (Sha Y. et al. Cancer Letters 2022, 545, 215832 and Wu J. et al. Cancers 2022, 14,5272 which have been added to references) we better discussed the relationship among PPAR β/δ, myeloma and immunomodulatory drugs in the Introduction section at page 2, lines 24-34.

Reviewer #2: There are several studies investigated Ppar Β/Δ role in tumor angiogenesis as the authors mentioned. Were there any findings which is inconsistent with the previous literature?

Action: The controversial role of PPAR β/δ in multiple myeloma has been examined in the Introduction section at page 2, lines 22-28.

Reviewer #2: I believe the limitations of the study should be mentioned.

Action: We thank the Reviewer for the useful comment. We have enriched the Discussion by examining the limitations of our study.

Reviewer 3 Report

This is a nice paper addressing in large the role of angiogenesis as an integral part of the microenvironment in asymptomatic MM vs MGUS. The authors were able to show that peroxisome proliferator-activated Receptor (PPAR β/δ) is expressed at higher levels in MM compared to MGUS and that prostaglandin I2 (PGI2) promotes the release of the PPAR β/δ ligand that stimulates angiogenesis in vitro, and that PPAR β/δ inhibition by a specific antagonist greatly impairs the PPAR mediated angiogenic effect.  The study is of course missing in vivo animal experiments which are mandatory. Furthermore, it will be nice to have more than 1 cell line, especially since the used line RPMI is not a line. I am not sure why the authors did not include symptomatic over MM. Also the combination studies they suggested in the discussion can be easily performed especially with other antiangiogenic compounds like IMIDs.

Author Response

We would like to thank very much the Reviewer for the constructive and helpful comments regarding our manuscript.

Reviewer #3: This is a nice paper addressing in large the role of angiogenesis as an integral part of the microenvironment in asymptomatic MM vs MGUS. The authors were able to show that peroxisome proliferator-activated Receptor (PPAR β/δ) is expressed at higher levels in MM compared to MGUS and that prostaglandin I2 (PGI2) promotes the release of the PPAR β/δ ligand that stimulates angiogenesis in vitro, and that PPAR β/δ inhibition by a specific antagonist greatly impairs the PPAR mediated angiogenic effect.  The study is of course missing in vivo animal experiments which are mandatory.

Action: We thank the Reviewer for the constructive criticism made to improve our manuscript quality. In vivo animal experiments have been included (page 14, figure 7).

Reviewer #3: Furthermore, it will be nice to have more than 1 cell line, especially since the used line RPMI is not a line.

Action: We used RPMI 8226 since it is a cell line of B lymphocytes isolated from a plasmacytoma patient and it is often used in immunology and immune system research.

Reviewer #3: I am not sure why the authors did not include symptomatic over MM.

Action: In our study we included newly diagnosed MM patients which met SLIM-CRAB criteria, are symptomatic and previously untreated. Furthermore, in the revised manuscript we have added a gene expression profiling study based on information acquired from the GSE9782 dataset of relapsed/refractory MM patients (page 14-15, figure 8).

Reviewer #3: Also the combination studies they suggested in the discussion can be easily performed especially with other antiangiogenic compounds like IMIDs.

Action: We thank the Reviewer for the comment that regards an interesting and important issue. The interaction between PPAR agonists/antagonists and IMIDs in MM has been also discussed in two very recent papers described in the Introduction at page 2, lines 24-34. Nevertheless, our study focuses on the role of PPAR β/δ in untreated MM patients and combination studies with antiangiogenic compounds like IMIDs would go beyond the scope of the study.

This limitation of our study is discussed in the Discussion section at page 16, lines 26-31.

Round 2

Reviewer 1 Report

The authors have satisfactorily answered all of my concerns. I have no additional comments.

Reviewer 3 Report

The authors  respond  partially to my comments and suggestions 

Still, they used only 1 cell line; Did not follow MM patients receiving treatments and try to correlate with clinical response